# Hypomineralized Teeth and Their Impact on Oral-Health-Related Quality of Life in Primary School Children

**DOI:** 10.3390/ijerph191610409

**Published:** 2022-08-21

**Authors:** Tim Reissenberger, Markus Ebel, Christian Klode, Christian Hirsch, Katrin Bekes

**Affiliations:** 1Independent Researcher, Odenthaler Straße 132, 51465 Bergisch Gladbach, Germany; 2Forum für Gesundheitswirtschaft Gemeinnützige GmbH, Am Wall 142, 28195 Bremen, Germany; 3School of Dentistry, Department of Paediatric Dentistry, University of Leipzig, Liebigstr. 12, 04103 Leipzig, Germany; 4Department of Paediatric Dentistry, University Clinic of Dentistry, Medical University of Vienna, Sensengasse 2a, 1090 Vienna, Austria

**Keywords:** molar–incisor hypomineralization, oral-health-related quality of life, Children Perceptions Questionnaire, pediatric dentistry

## Abstract

Background: Molar–incisor hypomineralization (MIH) has a strong negative effect on oral-health-related quality of life (OHRQoL). Malformed teeth can be hypersensitive, and the discoloration might affect children’s appearances, reducing their well-being. The purpose of the study was to investigate how hypomineralized incisors and molars differ in children’s perceived OHRQoL. Materials and Methods: 252 children aged 7–10 years old were included and subdivided into three equal groups (*n* = 84). Group A included children with asymptomatic molars and affected incisors. Group B included children presenting only affected molars. Group C was the control group, with children showing no MIH. All participants were asked to complete the German version of the Child Perceptions Questionnaire (CPQ-G8-10) to measure OHRQoL. Results: Participants in the posterior group showed a median total CPQ of 13.4 (±1.7), which was significantly higher than scores in the anterior and control group, which showed a median total CPQ of 8.4 (±1.4) and 4.2 (±0.7), respectively. Children in the posterior group suffered more from oral symptoms and functional limitations, whereas the anterior group dealt more with social and emotional well-being problems. Conclusions: The position of the MIH-affected teeth causes different influences on perceived OHRQoL.

## 1. Introduction

Molar–incisor hypomineralization (MIH) has gained importance in pediatric dentistry in the last few decades, mainly due to the increasing number of children who suffer from this enamel defect and the need for individual treatment. According to current research, MIH affects up to 14.2% of children worldwide. Among continents, South America currently has the highest prevalence at 18%. The prevalence in Europe is estimated to be 13.4%, which is about average. There are no differences in gender distribution [1]. The first descriptions of MIH in the literature date from 1987, when Koch et al. observed “idiopathic enamel defects” in a group of Swedish schoolchildren [2]. At the Congress of the European Academy of Pediatric Dentistry in Athens in 2000, the definition of MIH was established [3]. It has been described as a qualitative tooth structure defect leading to insufficient mineralization of the first molars that might also involve the incisors. The degree of discoloration and the form of the defect is highly individual. The extent varies in its appearance from small whitish–brownish opacities to massive post eruptive breakdowns with total loss of enamel. Such defects can arise on any kind of tooth, but MIH classically affects molars with/without the involvement of the incisors [4]. No systemic elicitor has been identified that could explain the ameloblasts’ impairment in the mineralization process. Therefore, it is assumed that a combination of synergetic agents around the date of birth blocks normal amelogenesis [5,6]. On the other hand, oral-health-related quality of life (OHRQoL) is a multilayered instrument that focuses on the subjective perception of a patient’s oral health, potential, functional limitations, and emotional and social well-being. MIH harms affected children’s OHRQoL, and this effect tends to worsen with the disease’s severity and might be more serious than in the case of large cavities or other serious oral diseases [7]. The porosity of these hypomineralized molars’ enamel reduces these teeth’s chewing capacity and often makes them very sensitive to thermal and mechanical stimuli. Post-eruptive enamel fractures promote the penetration of bacteria through exposed dentinal areas, which can result in chronic pulpitis [8]. Furthermore, the plaque-retentive surface and the teeth’s hypersensitivity decrease brushing ability, thereby negatively affecting oral hygiene and increasing the risk of cavities [9,10]. The possibility of profound anesthesia is greatly reduced because the enamel structure is malformed, making pain-free treatment more difficult. Therefore, complex therapy strategies place a considerable burden on the children [7]. In addition to the functional limitations caused by hypomineralized molars, discolored anterior teeth have a negative impact on OHRQoL. Extensive yellowish–brownish discoloration on the incisors’ buccal surfaces impairs young patients’ appearances. Some children feel so uncomfortable due to their structural defects that they even stop smiling to avoid teasing by classmates [11]. The Child Perceptions Questionnaire is a valid tool for measuring the extent of such impairments [12]. Several language versions exist making the questionnaire an instrument that can be applied in different countries [13].

This study is the first to evaluate how MIH-affected incisors influence OHRQoL compared to affected molars. Our hypothesis was that hypomineralized anterior teeth would cause a greater socio-emotional impact, as they play a crucial role in interactions with other people and are visible during smiling or speaking. On the other hand, molars are used more on a functional level during mastication, suggesting physical effects (e.g., pain when eating, drinking, etc.) rather than psychological effects.

## 2. Materials and Methods

### 2.1. Study Design

In this cross-sectional study, a consecutive sample of 252 schoolchildren aged 7 to 10, were evaluated on their OHRQoL in connection with MIH.

### 2.2. Subjects and Setting

The data collection source was Leo Löwenzahn, a private pediatric dentistry clinic in Bergisch Gladbach, North-Rhine-Westphalia, Germany. Patients were recruited from August 2020 to March 2021. To calibrate the diagnostics, standardized methods and clinical pictures were used in order to detect MIH. Potential teeth required a definable, marked off yellowish to brownish spot of more than 1 mm, separating it from the regular dental hard structure. Via the Schiff cold air sensitivity scale (SCASS) the degree of hypersensitivity was established. The air syringe attached to the dental unit, was held one second one cm away from the examined tooth with a pressure of 60 psi. The teeth were rated as 0 (no response), 1 (response but no removal request), 2 (removal request and patient movement away from the airflow), or 3 (removal request, leaning away from the stimulus and a feeling of pain). In a designated meeting, the calibrated lead examiner briefed the dental staff about the aims of the study so that they could look for potential participants. All patients aged 7–10, entering the dental clinic were examined by one of the six dentists. Only after confirmation by the calibrated lead examiner, patients were considered as participants. First, the dental status was evaluated and put into a data base. Under the same conditions, including dental unit, lighting, and airflow, a standardized diagnosis was possible. If patients fit the study criteria, they were provided with extensive instructions for the study. The study was approved by the ethics committee of the University of Leipzig (AZ: 152/19-ek).

To be included, the patients had to be between 7 and 10 years of age and present with a minimum of one MIH-affected tooth, either posterior or anterior with an asymptomatic posterior tooth, otherwise, they had to be completely healthy. Children, suffering from cavities, abscesses, or any pathological lesion in the oral cavity were excluded immediately. Restorations were also considered an exclusion criterion, since fillings might alter OHRQoL. Participants had to be free of any oral disease in the last six weeks, except for their MIH teeth, to form a segregated study pool. Furthermore, the children had to consent to the examination and needed to be able to state their feelings, meaning that they could complete the questionnaire on their own as much as possible. Patients undergoing orthodontic therapy had to be excluded as well due to the possible influences coming from orthodontic appliances instead of MIH. In addition, children with physical or psychological health issues and children who had problems expressing themselves were excluded because of possible distortion of the results. If the patient and caregiver agreed to participate in this study, they signed an agreement.

### 2.3. Sample Size

The children were subdivided into three categories based on tooth position.

Group A, anterior group: children with MIH-affected incisors or canines (with an asymptomatic MIH-affected molar also being present);Group B, posterior group: children with symptomatic MIH-affected molars (without incisors that impair OHRQoL); andGroup C, control group: participants who did not have any hypomineralized or otherwise affected teeth.

Since anterior MIH teeth by definition also present with a first molar affected by MIH, both tooth types were included in both groups in this study. In order to exclude any impairment of OHRQoL by a tooth of the other group, it was mandatory that the additional MIH tooth did not cause any symptoms or impairment for the patient. The individual severity of each hypomineralized tooth was also evaluated using Mathu-Muju and Wright’s classification method: low (I), moderate (II), or severe (III). To achieve a confidence interval of 95%, a sufficient sample size was calculated to be able to compare the three groups, using G*power Software (https://www.psychologie.hhu.de/arbeitsgruppen/allgemeine-psychologie-und-arbeitspsychologie/gpower (accessed on 18 June 2022)). While using an effect size *f* of 0.25 and an error probability of alpha (=0.05), this indicated that 84 participants per group were necessary for a sample size of 252 children.

### 2.4. Variables

The subscales of the German version of the CPQ-G8-10 and the total CPQ score were defined as the quantitative outcome variables, and the MIH position (i.e., anterior, posterior, or control) was defined as the independent group variable. Gender, age, and social status (according to Winkler and Stolzenberg [14]) were used as control variables. Patients rated their OHRQoL using the German version of the CPQ-G8-10 [13], which consists of 2 introductory questions and 25 main questions, all regarding the children’s subjective perception of their oral health. With the help of a “rate statement” form, the children and their parents had the opportunity to describe their individual feelings in a way that was statistically measurable. Because open-ended questions would have made evaluation rather difficult, we provided a Likert scale [15], allowing participants to answer each individual statement with 0, never; 1, once/twice; 2, sometimes; 3, often; or 4, very often.

Regarding the questions of the German CPQ-G8-10, the first two items operate as an overview of whether the child is satisfied with her/his general and oral health. The CPQ-8–10 comprises of four subscales, namely oral symptoms (5 items), functional limitations (5 items), emotional (5 items), and social wellbeing (10 items). A higher CPQ-G8-10 score indicates a lower OHRQoL. A maximum of 100 points indicates the worst possible OHRQoL, and zero points indicates the absence of any problems.

### 2.5. Data Sources and Measurements

After ensuring that the patient and caregiver understood the study’s format, written consent to provide the questionnaire was requested. After collection, all information was saved in a Microsoft Access 2019 database to establish a standardized collection. If a participant did not fully complete the questionnaire, she/he was excluded. To maintain data privacy and ensure anonymity, each patient was given a random number.

### 2.6. Bias

A potential source of influence is the caregivers who accompanied the young patients on the day of their examination, but we took measures to minimize the exertion of influence, such as a dental assistant being nearby to explain the questionnaire and to watch the participants complete it. The caregiver may have been a mother, father, grandparent, or legal guardian, and that person’s influence might have led to statements different from the patient’s intended response.

### 2.7. Statistical Methods

The German version of the CPQ-G8-10 questionnaire functioned as the base for the calculation. Furthermore, the CPQ-G8-10 subcategories were added to allow for a detailed analysis of variance. Analysis of variance (ANOVA) was conducted to test for significant differences between groups (i.e., anterior, posterior, and control groups) concerning the CPQ measures. We controlled for sociodemographic variables, such as gender, age, and social status (high, middle, and low). The calculations yielded frequencies, means, *p*-values (considered significant if the *p*-value is <0.05), and 95% confidence intervals (CI).

## 3. Results

### 3.1. Participants

The participants’ gender distribution was 131 female and 121 male participants. The study population had a mean age of 8.3 years (±2 years), and most of the children were 7 years old (31%). Children younger than age 10 comprised 20% (50) of the study pool (Table 1). The majority of the children were of middle social status (56%), 16% were of social status, and 28% were of high social status. 85% of the children were Caucasian (Table 1).

### 3.2. Main Results

Of the 5833 examined teeth, 742 were diagnosed as being MIH-affected (Appendix A and Appendix B). Group A included slightly more teeth, with 409 (55%), and Group B included 333 teeth (45%). Almost all of the anterior MIH-affected teeth were permanent ones, and only 30 (7%) were hypomineralized deciduous teeth (Table 2). The posterior group displayed similar characteristics, although the distribution was more divergent, with 233 (70%) permanent and 100 (30%) deciduous teeth. The control group consisted of 933 deciduous teeth and 1000 permanent teeth, of which 962 were incisors and canines and 971 were molars (Table 2), respectively. In the anterior group, we assigned most of the teeth (81%) were assigned to the low-severity category and placed only a few (18%) in the second category. The posterior MIH teeth group had 138 (41%) teeth in the low-severity group, and we placed 195 (59%) teeth in the second and third severity categories (Table 2).

Testing the different groups against each other yielded significant results. Except for the oral symptoms (*p* = 0.141) and functional limitations (*p* = 0.113) anterior versus posterior subscales, all other groups that were differentiated against each other had a *p*-value of less than 0.05.

The posterior group versus the control groups showed significant results, as the CPQ values of the posterior group were invariably higher than the CPQ scores of the control group (Table 3). The same was found for the emotional and social well-being subscales of the anterior group, which were compared with these subscales of the control group.

Because the CPQ scores of Group A for oral symptoms (2.3 [±0.5]) and functional limitations (0.9 [±0.4]) were lower than the CPQ scores of the posterior group (6.1 [±0.7] and 4.4 [±0.7], respectively), the differentiation was also highly significant. In contrast, the opposite was true for the CPQ scores for emotional well-being (3.8 [±0.8]) and social well-being (1.4 [±0.4]) of the anterior group, as the posterior group had lower CPQ scores (2.6 [±0.4], 0.4 [±0.3]) (Table 3).

Overall, the posterior group had the highest total CPQ value of 13.4 (±1.7), compared with 8.4 (±1.4) in the anterior group and 4.2 (±0.8) in the control group (Table 3).

Table 4 shows an overview of the CPQ scores graded by the group. The control group had the lowest CPQ scores in general. Although the majority of this group (59%) showed CPQ scores of 1–5, 14 participants (17%) presented CPQ scores of 11–20 (Table 4). Regarding the children in the anterior group, the majority (30%) scored 6–10 points, and 24 (29%) scored 11–20 points. Children in the posterior group generally rated their OHRQoL the worst. Most children (61%) rated their OHRQoL with a total CPQ in the range of 11–30. The differences in CPQ scores were highly significant (*p* = 0.000; Table 4).

We found no significant differences between age, social status, and gender distribution. See Appendix C, Appendix D and Appendix E.

## 4. Discussion

Several studies have been published that measure MIH’s effects on children’s OHRQoL [16,17,18,19,20,21,22]. However, none of these studies showed to what extent individual tooth position alters children’s perceptions of their OHRQoL. We compared children whose quality of life was impaired by either hypomineralized molars or hypomineralized anterior teeth. Despite an intensive search in medical and scientific online databases, such as PubMed and Google Scholar, we found no relevant publications matching our criteria.

The mean age of all subjects was 8.3 years. Only the posterior teeth group was about half a year younger than the rest of the subjects, which may be because children with hypomineralized molars often require early treatment due to hypersensitivity. However, because we included only children with untreated teeth, these patients were therefore not included. For statistical reasons, we matched the control group, meaning that we interviewed the same number of children in each age group.

The gender distribution in the two MIH groups was nearly equal. This distribution is consistent with the current literature, which states that both sexes have an equal incidence of MIH [1,7,9,13,16]. The distribution of ethnicity and social status, following Winkler and Stolzenberg, was similar in all three groups [14].

The results showed that there are significant differences if we test the individual groups against each other. Especially in comparison with a healthy control group, the results showed that MIH significantly reduced the quality of life in the anterior and posterior groups. Hypomineralized molars, which are mainly used to grind food, may cause a painful chewing experience, resulting in significantly reduced OHRQoL [18,23,24,25,26,27,28]. A suggested diet would be one with rather low thermal or chemical stimuli, e.g., no ice-cold drinks or highly seasoned foods.

The opposite was shown for the emotional and social well-being subscales, as the anterior group had the highest scores in these subscales. This result confirmed our hypothesis that the esthetic impairment caused by severely discolored anterior teeth can lead to frustration in children because the anterior teeth are visible when smiling, speaking, or eating. Analysis of the questionnaires revealed that these children were ashamed of their teeth and sometimes even avoided smiling. The group with posterior teeth also showed impairments, with significant differences compared to the control group, especially in the social well-being subscale. Subjects in this group were more likely to be annoyed or frustrated by their painful teeth. Some children even reported difficulty in completing their homework.

In the overall CPQ results, the posterior group had the highest score. This suggests that diseased molars drastically reduce OHRQoL. The participants in the anterior group, on the other hand, still scored significantly higher than those in the control group, also indicating that their OHRQoL was also impaired. Many researchers have assessed the OHRQoL with the CPQ-G8-10, so the obtained data can be compared. Originally, only children aged 8–10 were examined with this instrument because they had to be skilled enough in reading to answer the questions without help. But here 7-year-old children who met the selection criteria of this study were also included, if they could understand and answer the questionnaire thoughtfully. To reach a sufficient number of participants and to generate statistically relevant statements, we needed to select these younger participants, since many children who visit the dentist at a more advanced age have already been treated for their hypomineralized teeth, drastically reducing the impairment in their quality of life.

In the posterior group, more than two-thirds of all hypomineralized teeth were permanent first molars, mainly because of the definition of MIH. We also found deciduous teeth affected by hypomineralized second primary molars (HSPM), as has occurred in numerous studies [9]. We only included subjects if they rated the negative impact on their OHRQoL as non-existent. These teeth exhibited no hypersensitivity, nor did they show any loss of substance. In the anterior group, the distribution of the diseased teeth looked very different. About half of all hypomineralized teeth were permanent incisors or canines. As previously mentioned, by definition, at least one 6-year-old molar must also be hypomineralized [3]. However, only children whose permanent molars were not symptomatic were included in this group, which means these teeth did not show any loss of substance or present with a physiological reaction to cold stimuli.

The evaluation focus of the data was the analysis of the CPG-G8-10 because significant differences between the three groups were found. As expected, the control group had the lowest total CPQ score because these children suffered from no oral diseases. The literature in this regard is relatively sparse because most studies have concerned only the documentation of sick children. Our results come closest to the data from the Austrian CPQ-G8-10 study by Bekes et al. [13], thus providing an exception to the current literature. Their mean CPQ score was 6.5 (±6.8) for children without any oral diseases in that particular age group, which was slightly higher than in our study.

This study’s data is only partially comparable with that of other studies due to our specific grouping criteria, but all studies in this field confirm MIH’s negative influence on OHRQoL. The involvement of MIH decreases OHRQoL in general, as a systemic review of international papers by Jävelik et al. suggested [19]. Joshi et al. applied the CPQ to hypomineralized teeth, and the children had a total CPQ score of 13.9 (±8.9) [6]. This value is significantly higher than in our data. Our selection criteria might explain this finding. On one hand, our participants had either symptomatic hypomineralized molars or symptomatic hypomineralized frontal teeth, suggesting that there were no potential additive effects. According to our observations, the more severe the MIH in affected molars, the more involvement of frontal teeth we found. Therefore, we had to exclude children severely with severely hypomineralized teeth based on the inclusion criteria. Michaelis et al. examined participants mainly with regard to severity [7], showing that the total CPQs of subjects with mild cases of MIH were only slightly greater than those of our healthy control group. In contrast, children with severe MIH had values almost twice as high as those of the children in this study’s posterior group, which can also be attributed to additive effects of the frontal teeth in conjunction with the posterior teeth.

The present study has several strengths and some limitations. The approach we took in this study, where a strict separation was made between the three groups, has not been taken before. The MIH anterior teeth group included patients with MIH molars, but only if these molars were asymptomatic. If there was a possibility of altered OHRQoL due to the molars, these children were not included because only one group of teeth (anterior or posterior) should be examined at a time in this study. In contrast, the hypomineralized posterior teeth group also included MIH incisors in the study, but only if patients did not report any problems in these teeth. A large number of participants is necessarily required to implement such a new approach; we were able to realize our approach by conducting the study in a large private dental clinic specializing in pediatric dentistry, where it was possible to obtain a sufficient number of children willing to participate. The total number of participants was further increased by including seven-year-old children. Because the selection criteria were highly selective, good standardization was possible. An already established valid questionnaire (CPQ-G8-10) was chosen so we could compare our results with those of other studies in the literature. The selected age group limited the participant pool, which is why we included seven-year-old children. The CPQ-G8-10 can be considered as a broad measuring instrument, but it can never show all the limitations a young individual might have due to their MIH-affected teeth. We attempted to objectify the different groups as much as possible, but there still remains the possibility of invariant influences: eating habits could influence perceived OHRQoL, as could parental hygiene education. Social class could also alter subjective OHRQoL, as could socioeconomic opportunities to have the disease treated. Parental influence on the child cannot be ruled out either when discussing the individual statements. Each child perceives his or her OHRQoL somewhat differently, as it is a subjective concept, which could introduce bias into the underlying data. In addition, future studies on this topic could include even larger numbers of analyzed teeth. However, the strict selection of the study participants in conjunction with the separation into teeth groups led to significant results, which proves that anterior and posterior MIH-affected teeth alter the quality of life in different ways. As anterior MIH teeth are visible during human interaction, children tend to experience more socio-emotional problems. On the contrary posterior MIH teeth are used during mastication, therefore leading to hypersensitivity and limitations in function.

## 5. Conclusions

This study suggests that the position of an MIH-affected tooth has a strong impact on perceived OHRQoL. Hypomineralized molars caused the most severe oral symptoms and functional limitations. Hypomineralized incisors had the largest negative effect on OHRQoL in the emotional well-being and social well-being subscales. Further research is necessary to confirm and extend this study’s results.

## Figures and Tables

**Table 1 ijerph-19-10409-t001:** Summary of Sample Data (*n* = 252 patients).

	Anterior-Group **	Posterior-Group **	Control-Group
	% (*n*)	% (*n*)	% (*n*)
**Gender**			
Male ♂	48.8 (41)	45.2 (38)	50.0 (42)
Female ♀	51.2 (43)	54.8 (46)	50.0 (42)
**Social status ***			
Low	17.9 (15)	14.3 (12)	15.5 (13)
Medium	55.9 (47)	58.3 (49)	53.6 (45)
High	26.2 (22)	27.4 (23)	31.0 (26)
**Ethnicity**			
Caucasian	83.3 (70)	85.7 (72)	84.5 (71)
Non-Caucasian	16.7 (14)	14.3 (12)	15.5 (13)
**Age**			
Mean	8.6 years	7.8 years	8.5 years
7 years	20.2 (17)	46.4 (39)	25.0 (21)
8 years	27.4 (23)	32.1 (27)	25.0 (21)
9 years	27.4 (23)	11.9 (10)	25.0 (21)
10 years	25.0 (21)	9.5 (8)	25.0 (21)

* Social status was determined according to the classification by Winkler & Stolzenberg [14]. ** Anterior-Group was defined as any affected hypomineralized incisor or canine, which might have also included a asymptomatic MIH molar. Posterior-Group was defined as any molar, which was affected by MIH. Control-Group patients had no MIH teeth at all.

**Table 2 ijerph-19-10409-t002:** Affected MIH Teeth in the Anterior- and Posterior-Group.

	All 742 Affected Teeth	
	Anterior-Group *	Posterior-Group *
	% (*n*)	
**Anterior teeth (incisors and canines) 55.1% (409)**		
Deciduous	3.8 (8)	100 (30)
Permanent	96.2 (203)	0 (0)
**Posterior teeth (deciduous molars, premolars, molars), 44.9% (333)**		
Deciduous	11.1 (22)	23.1 (70)
Permanent	88.9 (176)	76.9 (233)

Abbreviations: MIH = molar incisor hypomineralization. * Anterior-Group was defined as any affected hypomineralized incisor or canine, which might have also included a asymptomatic MIH molar. Posterior-Group was defined as any molar, which was affected by MIH. Control-Group patients had no MIH teeth at all.

**Table 3 ijerph-19-10409-t003:** Mean and confidence intervals for scores in each CPQ subscale of the Anterior-, Posterior- and Control-Group.

CPQ Subscales	Anterior-Group *		Posterior-Group *		Control-Group		Anterior vs. Control	Posterior vs. Control	Anterior vs. Posterior
	Mean CPQ	CI 95%	Mean CPQ	CI 95%	Mean CPQ	CI 95%	DIFF Test	DIFF Test	DIFF Test
						Lower Bound	*p*-Value	*p*-Value	*p*-Value
Oral symptoms	2.3	1.9–2.8	6.1	5.4–6.7	3.1	2.6–4.7	0.141	0.0	0.0
Functional limitations	0.9	0.5–1.2	4.4	3.7–5.1	0.3	0.2–0.5	0.113	0.0	0.0
Emotional wellbeing	3.8	3.0–4.5	2.6	2.2–3.0	0.6	0.5–0.8	0.0	0.0	0.001
Social wellbeing	1.4	1.1–1.8	0.4	0.2–0.7	0.1	0.0–0.2	0.0	0.0	0.0
Total CPQ	8.4	7.0–9.8	13.4	11.7–15.1	4.2	3.5–5.0	0.0	0.0	0.0

Considered as significant if *p*-value < 0.05. Abbreviations: CPQ = Child Perceptions Questionnaire, CI = Confidence Interval, DIFF Test= Paired difference test. * Anterior-Group was defined as any affected hypomineralized incisor or canine, which might have also included a asymptomatic MIH molar. Posterior-Group was defined as any molar, which was affected by MIH. Control-Group patients had no MIH teeth at all.

**Table 4 ijerph-19-10409-t004:** Range of the Total-CPQ-Scores.

	Anterior-Group *	Posterior-Group *	Control-Group
CPQ Scores	% (*n*)	% (*n*)	% (*n*)
0	10.7 (9)	4.8 (4)	11.1 (9)
1–5	26.2 (22)	19.0 (16)	59.3 (48)
6–10	29.8 (25)	14.3 (12)	23.5(19)
11–20	28.6 (24)	42.9 (36)	6.2 (5)
>31	4.8 (4)	19.1 (16)	0
max CPQ	27	35	15

Abbreviations: CPQ = Child Perceptions Questionnaire. * Anterior-Group was defined as any affected hypomineralized incisor or canine, which might have also included a asymptomatic MIH molar. Posterior-Group was defined as any molar, which was affected by MIH. Control-Group patients had no MIH teeth at all.

## Data Availability

The datasets of this article are available from the corresponding author upon reasonable request.

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
