# Peer review of "Hypomineralized Teeth and Their Impact on Oral-Health-Related Quality of Life in Primary School Children"

_ijerph, 2022, doi:10.3390/ijerph191610409_

Round 1
Reviewer 1 Report
Dear Authors,
The study is well written and could merit to be published after a major revision. I list here my suggestions.
TITLE
I think that the title could be simpler and more endearing as for example:
HYPOMINERALIZED TEETH AND THEIR IMPACT ON THE ORAL-HEALTH-RELATED QUALITY OF LIFE IN PRIMARY SCHOOL CHILDREN
ABSTRACT
Ok, well done
INTRODUCTION
please correct the bibliographic citation format according to the journal rules
Well done, you could also focus on the geographical distribution of MIH as - Dave M, Taylor G. Global prevalence of molar incisor hypomineralisation. Evid Based Dent. 2018 Oct;19(3):78-79. doi: 10.1038/sj.ebd.6401324. PMID: 30361661 – or others, in order to highlight how MIH can vary / not vary according to ethnicity and local diet
M&M
Well done
DISCUSSION and CONCLUSIONS
I think that in this study it would be useful to consider the eating habits of the patients, perhaps suggesting a diet to be followed and carrying out an oral hygiene diet, to make the sample more homogeneous. If we think about the different eating habits of children in the age group considered, the economic possibilities, the hygiene education and the social class (in which it is perhaps worse to have an MIH on an incisor than on a molar), the groups could not be comparable at best. Perhaps these aspects could be mentioned as limitations of the study. However, it remains an interesting and stimulating study, as you said, to continue to deepen the subject, so the conclusions are appropriate.
Best Regards
Author Response
Dear Editor
We thank the reviewers for their thoughtful and helpful comments and suggestions. Please find enclosed our detailed point-by-point response.
We have revised the manuscript accordingly, indicating all changes in red. We feel our manuscript has substantially improved as a direct result of these comments and hope the revised version is suitable for publication
Remark #1: TITLE: I think that the title could be simpler and more endearing as for example: HYPOMINERALIZED TEETH AND THEIR IMPACT ON THE ORAL-HEALTH-RELATED QUALITY OF LIFE IN PRIMARY SCHOOL CHILDREN
Response Remark #1: We thank the reviewer for this suggestion and have changed the title accordingly.
Remark #2: INTRODUCTION: please correct the bibliographic citation format according to the journal rules.
Response Remark #2: Thank you for this comment. We have formatted the citations correctly now.
Remark #3: Well done, you could also focus on the geographical distribution of MIH as - Dave M, Taylor G. Global prevalence of molar incisor hypomineralisation. Evid Based Dent. 2018 Oct;19(3):78-79. doi: 10.1038/sj.ebd.6401324. PMID: 30361661 – or others, in order to highlight how MIH can vary / not vary according to ethnicity and local diet
Response Remark #3:Thank you for reminding us regarding this meta-analysis. We have added the reference at the beginning of our introduction.
Revised text: page 1, lines 20-23 „According to current research, MIH affects up to 14.2% of children worldwide. Among continents, South America cur- rently has the highest prevalence at 18%. The prevalence in Europe is estimated to be 13.4%, which is about average. There are no differences in gender distribution [1]
”
Remark #4: DISCUSSION and CONCLUSIONS: I think that in this study it would be useful to consider the eating habits of the patients, perhaps suggesting a diet to be followed and carrying out an oral hygiene diet, to make the sample more homogeneous.
Response Remark #4: We thank the reviewer for his/her thoughtful comment. We have incorporated a diet suggestion eg no ice-cold beverages, no food which is too seasoned.
Revised text: page 7, lines 220-222 “Molars, which are mainly used to grind food, may cause a painful chewing experience if affected by MIH, resulting in significantly reduced OHRQoL [18, 24-29]. A suggested diet would be one with rather low thermal or chemical stimuli, e.g., no ice-cold drinks or highly seasoned foods.“
Remark #5: If we think about the different eating habits of children in the age group considered, the economic possibilities, the hygiene education and the social class (in which it is perhaps worse to have an MIH on an incisor than on a molar), the groups could not be comparable at best.
Response Remark #5: Thank you for suggesting further limitations. We agree on this point and have added these limitations as well.
Revised text: page 8, lines 295-302 „We attempted to objectify the different groups as much as possible, but there still remains the possibility of invariant influences: eating habits could influence perceived OHRQoL, as could parental hygiene education. Social class could also alter subjective OHRQoL, as could socioeconomic opportunities to have the disease treated. Parental influence on the child cannot be ruled out either when discussing the individual statements. Each child perceives his or her OHRQoL somewhat differently, as it is a subjective concept, which could introduce bias into the underlying data.“
Reviewer 2 Report
line 55 to 59 of the introduction need not be written, The need for the research or novelty can be elaborated in the introduction earlier and not towards the end, the intro section can end with the intro or hypothesis
In the sample size calculation and additional 10% sample for drop out should have been considered
in the methods section, line 124-127 can be moved to the discussion section
The discussion is sufficient, however limitations and possible future studies need to be included towards the end
Ethics approval code shoud be included in Human research ethics approval statement
The article can be accepted after the above comments have been addressed, Thank you
Author Response
Dear Editor
We thank the reviewers for their thoughtful and helpful comments and suggestions. Please find enclosed our detailed point-by-point response.
We have revised the manuscript accordingly, indicating all changes in red. We feel our manuscript has substantially improved as a direct result of these comments and hope the revised version is suitable for publication
Remark #1: line 55 to 59 of the introduction need not be written, The need for the research or novelty can be elaborated in the introduction earlier and not towards the end, the intro section can end with the intro or hypothesis.
Response Remark #1: Thank you very much for this comment. We agree and have changed the introduction. You are right, it has been changed plus I tried to clarify the hypothesis a bit further.
Revised text: page 1, lines 55-60 „This study is the first to evaluate how MIH-affected incisors influence OHRQoL compared to affected molars. Our hypothesis was that hypomineralized anterior teeth would cause a greater socio-emotional impact, as they play crucial role in interactions with other people and are visible during smiling or speaking. On the other hand, molars are used more on a functional level during mastication, suggesting physical effects (e.g., pain when eating, drinking, etc.) rather than psychological effects.“
Remark #2: In the sample size calculation and additional 10% sample for drop out should have been considered
Response Remark #2: Thank you very much for this comment. We would like to explain the setting in the dental practice that was involved in this study. When a new patient enters the dental practice, he/she is immediately informed about the CPQ8-10 and if the patient agrees, he/she is asked to fill it out. Therefore, the participants selection pool was way higher than the actual analyzed number.
In order to get a sufficient sample size, we used a software called G*power analysis, which is a tool providing a study pool number where the sample drop is already included in the calculation.
At the end, 36 participants, who would have matched the criteria, have been sorted out for various reasons, such as incomplete questionnaires.
Remark #3: in the methods section, line 124-127 can be moved to the discussion section
Response Remark #3: We thank the reviewer for this suggestion and have moved this part under the limitations section in the discussion. The text can now be found on page 7, line 293-296.
Revised text: page 7, line 299-300 “Parental influence on the child also cannot be ruled out when discussing the individual statements.”
Remark #4: The discussion is sufficient, however limitations and possible future studies need to be included towards the end
Response Remark #4: We absolutely agree with the reviewer and have added further limitations.
For example, suggesting greater study pools than just 742 teeth.
Revised text: page 7, lines 292-302 “The selected age group limited the participant pool, which is why we included 7-year-old children. The CPQ-G8-10 can be considered as a broad measuring instrument, still it can never show all the limitations a young individual might have due to their MIH-affected teeth. An approach was taken to objectify the different groups as much as possible, but there still remains the possibility of invariant influences: Eating habits could influence perceived OHRQoL, as could parental hygiene education. Social class could also alter subjective OHRQoL, as could socioeconomic opportunities to have the disease treated. Parental influence on the child also cannot be ruled out when discussing the individual statements. Each child perceives his or her OHRQoL somewhat differently, as it is a subjective concept, which could introduce bias into the underlying data. In addition, future studies on this topic could include even larger numbers of teeth analyzed.”
Remark #4: Ethics approval code shoud be included in Human research ethics approval statement
Response Remark #4: Thank you very much for this comment. We have added the ethics approval code.
Revised text: page 2, line 73-74 „The study was approved by the ethics committee of the University of Leipzig (AZ: 152/19-ek).“
Reviewer 3 Report
It would be helpful for referees (who as you know perform this task free of charge and as a service to the scientific community) to not make them search for tables in 'additional documents'. Please add them to the text!
The text contains a lot of germanisms. Proband in English has a slightly diffrent meaning.
A better formulated research hypothesis can contribute to better reading.
In my opinion, there is a high risk of bias in the data: As shown in table 2 (which, as mentioned before, I had to dig up from supplementary files), you also have MIH-affected posterior teeth in the 'anterior' group and vice versa. As untreated lesions, abscesses or other pathological processes can also impact quality of life, did you detect differences in caries in the primary or permanent dentition between the groups?
The statistical methods are ill described and from all I can conclude from the tables is that a rather simplistic approach was followed. It is a pity that all the trouble taken in data collection is invalidated by insufficient data processing.
In the light of the above, the discussion should be rewritten and above all, be shortened significantly to point out the interesting points and not drowning this in a lot of less relevant text.
Please find notes on unclear expressions (annotated 'XP') and other suggestions for text amendments in the attached file.

Author Response
Dear Editor
We thank the reviewers for their thoughtful and helpful comments and suggestions. Please find enclosed our detailed point-by-point response.
We have revised the manuscript accordingly, indicating all changes in red. We feel our manuscript has substantially improved as a direct result of these comments and hope the revised version is suitable for publication
Remark #1: It would be helpful for referees (who as you know perform this task free of charge and as a service to the scientific community) to not make them search for tables in 'additional documents'. Please add them to the text!
Response Remark #1: Thank you very much for this note. We have incorporated the tables that were added as additional documents in the manuscript.
Page 4, line 158; page 5, line 171; page 6, line 187; page 6, line 195; appendices are found on page 9-11, line 330
Remark #2: The text contains a lot of germanisms. Proband in English has a slightly diffrent meaning.
Response Remark #2: Thank you very much for this comment. We have changed "probands" to "participants" and "caries" to "cavities”. Furthermore, the paper was lectured again with the specific request to eliminate any Germanism.
Remark #3: A better formulated research hypothesis can contribute to better reading.
Response Remark #3: Thank you the reviewer for his/her comment, we agree and have changed the text.
Revised text: page 1, lines 55-60 „This study is the first to evaluate how MIH-affected incisors influence OHRQoL compared to affected molars. Our hypothesis was that hypomineralized anterior teeth would cause a greater socio-emotional impact, as they play crucial role in interactions with other people and are visible during smiling or speaking. On the other hand, molars are used more on a functional level during mastication, suggesting physical effects (e.g., pain when eating, drinking, etc.) rather than psychological effects.“
Remark #4: In my opinion, there is a high risk of bias in the data: As shown in table 2 (which, as mentioned before, I had to dig up from supplementary files), you also have MIH-affected posterior teeth in the 'anterior' group and vice versa. As untreated lesions, abscesses or other pathological processes can also impact quality of life, did you detect differences in caries in the primary or permanent dentition between the groups?
Response Remark #4: We thank the reviewer for the comment. Per definition MIH anterior teeth are per EAPD criteria always in combination with an affected MIH posterior tooth. Our approach to deal with this issue, was to select strong selection criteria, meaning none of the hypomineralized molars in anterior group did alter the perceived OHRQoL as they only showed mild forms (showing only opacities) at all and vice versa. We also used the the Schiff sensitivity score, to make sure to exclude possible sensitivities that might interfere with quality of life.
Vice versa, the MIH incisors in the MIH posterior group did not cause any problems for the participant. MIH in affected incisors of this group had to be so small, that patients did not even knew about their diagnosis before, and the hypersensitivity test also had to be negative.
To make this more clear in the manuscript, we have added some lines.
Revised text: page 2, lines 75-86 “To be included, the patients had to be between 7 and 10 years of age and present with a minimum of one MIH-affected tooth, either posterior or anterior with an asymptomatic posterior tooth, otherwise they had to be completely healthy. Children, suffering from cavities, abscesses or any pathological lesion in the oral cavity were excluded immediately. Furthermore, children had to consent to the examination and needed to be able to state their feelings, meaning that they could complete the questionnaire on their own as much as possible. Patients undergoing orthodontic therapy had to be excluded as well due to the possible influences coming from orthodontic appliances instead of MIH. In addition, children with physical or psychological health issues and children who had problems expressing themselves were excluded because of possible distortion of the results.
Remark #5: The statistical methods are ill described and from all I can conclude from the tables is that a rather simplistic approach was followed. It is a pity that all the trouble taken in data collection is invalidated by insufficient data processing.”
Response Remark #5: Thank you for highlighting it. To address your remarks, the MM has been heavily revised. The statistical methods were made more precise. The results and discussion section was rewritten to delete irrelevant parts.
Page 3, line 61-150
Remark #6: In the light of the above, the discussion should be rewritten and above all, be shortened significantly to point out the interesting points and not drowning this in a lot of less relevant text.
Response Remark #6: We thank the reviewer for this note. We have restructured the whole discussion part and have deleted irrelevant parts.
Revised text: page 7, line 199-308
Remark #7: Please find notes on unclear expressions (annotated 'XP') and other suggestions for text amendments in the attached file.
Response Remark #7: Thank you very much for your comments.
- The first address was doubled. Now it is corrected.
- The term "staff members recruiting patients" was changed into “In a designated meeting, the lead examiner briefed the dental staff about the aims of the study so that they could look for potential participants.” (page 2, lines 69-70)
- We have clarified the inclusion/exclusion criteria and describe them more precisely now (page 2, lines 75-87)
- The reference quotation that was doubled was corrected (page 11, line 339)
Reviewer 4 Report
Dear Authors,
Congratulations on your study. Let me suggest few corrections.
First of all please check the Mdpi instructions for the authors how to prepare a manuscript- the citation of the references should be placed in brackets []; the references need to be formated ; provide the author's affiliation with regard to the instructions;
Please divide the section material and methods into subsections, its gonna be easier to follow that section
The number and date of the approval of the study by the Ethics Committee should be provided
Author Response
Remark #1: First of all please check the Mdpi instructions for the authors how to prepare a manuscript- the citation of the references should be placed in brackets []; the references need to be formated ; provide the author's affiliation with regard to the instructions;
Response Remark #1: Thank you very much for this note. We have corrected this.
Remark #2: Please divide the section material and methods into subsections, its gonna be easier to follow that section
Response Remark #2: Thank you very much for this valuable comment. We have divided the materials and methods section in further subsections now. These are: study design, subjects and settings, sample size, variables, data sources and measurements, bias and statistical methods.
Remark #3: The number and date of the approval of the study by the Ethics Committee should be provided
Response Remark #3: Thank you very much for this comment. We have added the number and date of the approval of the study by the Ethics Committee.
Revised text: page 2, line 73-74 „The study was approved by the ethics committee of the University of Leipzig (AZ: 152/19-ek).“
Round 2
Reviewer 1 Report
Dear Authors,
I congratulate for the effort in improving the manuscript.
I think that now it could be suitable for publication.
Best Regards
Author Response
We thank the reviewer for their thoughtful and helpful comments and suggestions.
Reviewer 3 Report
The manuscript has now become much clearer bur it still bears an important source of bias being the selection criteria for the "anterior" and "posterior" group.
As the authors state in line 130 ff, molars should be "asymptomatic" in the anterior and "symptomatic" in the posterior group.
In the contrasting assumption, all incisors in the posterior group would be asymptomatic.
What then with the incisors in the "posterior" group? Were these asymptomatic as well? It is shown that they have a low grade of MIH (I) but what with the grade I incisors from the "anterior" group?
Did these contibute to the OHQuOL reduction or not?
This makes case selection a bit of a self-fulfilling prophecy regarding the (not very clearly formulated) work hypothesis.
I presume that the authors aimed at a kind of "case-control" model, however then in the wrong direction.
However, as the clinical and questionnaire data are available, a more comprehensive statistical analysis should be able to elucidate this as a covariate and evaluate its association with OHQuOL data.
The conclusions may not invalidate the present working hypothesis but reduce the risk of bias.
Further detailed questions:
Participants were selected as being "free of cavities". Does this include or exclude restaurations?
Were the examiners calibrated in some or other way for detection and grading of MIH? So yes, how? How was the presence of symptomes graded and recorded?
Was this reproducible?
Stating the number of teeth in a table is of secundary importance as these are no indiependent data, related to a patient.
Author Response
We thank the reviewers for their thoughtful and helpful comments and suggestions. Please find enclosed our detailed point-by-point response.
We have revised the manuscript accordingly, indicating all changes in red. We feel our manuscript has substantially improved as a direct result of these comments and hope the revised version is suitable for publication
Remark #1:
The manuscript has now become much clearer bur it still bears an important source of bias being the selection criteria for the "anterior" and "posterior" group.
As the authors state in line 130 ff, molars should be "asymptomatic" in the anterior and "symptomatic" in the posterior group.
In the contrasting assumption, all incisors in the posterior group would be asymptomatic.
What then with the incisors in the "posterior" group? Were these asymptomatic as well? It is shown that they have a low grade of MIH (I) but what with the grade I incisors from the "anterior" group?
Did these contribute to the OHQuOL reduction or not?
Response Remark #1:
We thank the reviewer for this comment. The EAPD defines MIH as a condition in which at least one first permanent molar must be hypomineralized - with or without incisor involvement. Therefore, MIH-affected molars must be included in the anterior group. To counteract any involvement of hypomineralized molars, we made sure that these teeth were only grade I (according to the MIH classification of Mathu-Muju and Wright). This means no hypersensitivity and no functional problems. Hypersensitivity was categorized using SCASS (Schiff cold air sensitivity scale). Teeth were examined with the syringe attached to the dental unit, which was held one centimeter from the examined tooth for one second at a pressure of 60 psi. The child's response was scored as 0 (no response), 1 (response but no request for removal), 2 (request for removal and moving the patient away from the air stream), or 3 (request for removal, leaning away from the stimulus and feeling pain). SCASS was applied to all affected and unaffected first permanent molars. This comparison revealed no differential effects on their OHRQoL.
The posterior group did not include any hypomineralized central and lateral incisors at all. Twenty participants in whom only the deciduous canines were affected remained in the study pool. These children were included in the study because we were able to rule out OHRQoL limitations for the child due to the canines being affected by MIH. This was done using the same method, to exclude hypersensitivity and functional problems. Furthermore, it was asked if the child or caregiver had ever noticed their MIH spot on the canines. Most of these MIH spots tended to be on the distal side of the tooth, making them particularly difficult to detect - thus the children indicated that they had not noticed the hypomineralized canine.
Since your comment is acknowledged, we examined for potential bias concerning unobserved relations between anterior and posterior conditions. We selected all anterior teeth within the posterior group, and all cases with at least one MIH grade. This resulted in 20 patients within the posterior group. Then we tested for significant differences of the CPQ and CPQ subscales between this patient group and 64 patients of the posterior group with no MIH grade. The ANOVA test revealed only minor differences which were all non-significant.
Therefore, we are confident that the groups in our dataset were differentiated widely enough to not affect the statistics and to eliminate any bias.
We have included the tables as appendices. Please see the attached analysis (MIH posterior group excerpt TR.pdf).
____________________________
Remark #2:
Participants were selected as being "free of cavities". Does this include or exclude restorations?
Response Remark #2:
We thank the reviewer for this comment. Restorations were considered as an exclusion criteria. We have added it to the manuscript.
Page 2, line 88-89: „Restorations were also considered an exclusion criteria, since fillings might alter the OHRQoL.“
____________________________
Remark #3:
Were the examiners calibrated in some or other way for detection and grading of MIH? So yes, how?
Response Remark #3:
We appreciate this comment from the reviewer. The dental clinic where the study was conducted is "GCP" certified. The head dentist at the dental clinic is experienced in this particular area of research, having conducted several MIH studies that have been published.
When one of the dentists encountered a potential participant, a new appointment was made for that child. Only the calibrated lead examiner T.R. then confirmed the presence of MIH via the EAPD criteria and provided the grade (Mathu-Muju and Wright classification). The SCASS test was used to establish the degree of hypersensitivity. The stimulus was applied to each set of teeth, whether hypomineralized or not. If the hypomineralized teeth responded more strongly to the stimulus than the non-hypomineralized teeth, hypersensitivity was detected.
The lead examiner was calibrated with the assistance of a specialist who is a trained DAJ (Deutsche Arbeitsgemeinschaft für Jugendzahnpflege) pediatric dentist. As a statistical target, a kappa value greater than 0.81 was considered sufficient to correctly diagnose MIH.
The data set used for training is included as an appendix (Training TR.pdf).
This was not fully apparent in the manuscript, so we have corrected it accordingly.
Page 2, line 68-72: „To calibrate the diagnostics, standardized methods and clinical pictures were used in order to detect MIH. In a designated meeting, the calibrated lead examiner briefed the dental staff about the aims of the study so that they could look for potential participants. Only after a confirmation by the calibrated lead examiner, patients were considered as participants.“
____________________________
Remark #4:
How was the presence of symptoms graded and recorded?
Was this reproducible?
Response Remark #4:
We thank the reviewer for this comment. The SCASS was used to determine hypersensitivity. Each hypomineralized tooth was recorded separately to compare the teeth with their non-hypomineralized molars. Several outcomes were recorded: 0 (no response), 1 (response but no request for removal), 2 (request for removal and patient movement away from the airstream), or 3 (request for removal, leaning away from the stimulus, and pain sensation).
To create a standardized, reproducible classification system, we used Mathu-Muju and Wright's classification. Once the lead examiner made the correct classification, it was entered directly into our database.
|
Mild |
Moderate |
Severe |
Crown appearance |
Demarcated opacities in non-stress-bearing area of molar |
Intact atypical restoration present |
Posteruptive enamel breakdown present |
Enamel loss |
Isolated opacities |
Occlusal/incisal third of teeth without initial posteruptive enamel breakdown |
Posteruptive enamel breakdown on erupting tooth that can be rapid |
Caries |
No caries associated with affected enamel |
Posteruptive enamel breakdown/caries limited to one or two surfaces without cuspal involvement |
Often develop widespread caries associated with affected enamel |
Sensitivity |
Normal dental sensitivity |
Usually normal dental sensitivity |
Usually, history of dental sensitivity |
Esthetics |
Usually not an issue |
Parents often express concern |
Parents typically concerned |
(Mathu-Muju K, Wright JT (2006) Diagnosis and treatment of molar incisor hypomineralization. Compend Contin Educ Dent 27:604–610)
Page 3, line 104-106: „The individual severity of each hypomineralized tooth was also evaluated using Mathu-Muju and Wright’s classification method: low (I), moderate (II), or severe (III).“
Page x, line x-x: „To calibrate the diagnostics, standardized methods and clinical pictures were used in order to detect MIH. Potential teeth required a definable, marked off yellowish to brownish spot of more than 1mm, separating it from the regular dental hard structure. Via the Schiff cold air sensitivity scale (SCASS) the degree of hypersensitivity was established. The air syringe attached to the dental unit, was hold one second one cm away from the examined tooth with a pressure of 60 psi. The teeth were rated as 0 (no response), 1 (response but no removal request), 2 (removal request and patient movement away from the airflow), or 3 (removal request, leaning away from the stimulus and a feeling of pain). In a designated meeting, the calibrated lead examiner briefed the dental staff about the aims of the study so that they could look for potential participants. All patients, aged 7-10, entering the dental clinic were examined from one of the six dentists. Only after a confirmation by the calibrated lead examiner, patients were considered as participants.“
____________________________
Remark #5:
Stating the number of teeth in a table is of secondary importance as these are no independent data, related to a patient.
Response Remark #5:
We appreciate this remark. The table of teeth data functions only as an overview of our data pool.
____________________________
Remark #6:
This makes case selection a bit of a self-fulfilling prophecy regarding the (not very clearly formulated) work hypothesis.
I presume that the authors aimed at a kind of "case-control" model, however then in the wrong direction.
However, as the clinical and questionnaire data are available, a more comprehensive statistical analysis should be able to elucidate this as a covariate and evaluate its association with OHQuOL data.
The conclusions may not invalidate the present working hypothesis but reduce the risk of bias.
Response Remark #6:
We thank the reviewer for this statement.
We ran some of the statistics again to see if this would change anything in the significant results.
It did not affect the outcome of the study. We noted it to do it differently for the next study, as we find the reviewer's comment useful.
Reviewer 4 Report
The manuscript has been approved. In my opinion it is suitable for the publication.
Author Response
We thank the reviewers for their thoughtful and helpful comments and suggestions.